# Decoupled Cross-Modal Transformer for Referring Video Object Segmentation

**DOI:** 10.3390/s24165375

**Published:** 2024-08-20

**Authors:** Ao Wu, Rong Wang, Quange Tan, Zhenfeng Song

**Affiliations:** 1School of Information and Cyber Security, People’s Public Security University of China, Beijing 100038, China; 15262275428@163.com (A.W.); tanquange@126.com (Q.T.); songzhenfeng@ppsuc.edu.cn (Z.S.); 2Key Laboratory of Security Prevention Technology and Risk Assessment of Ministry of Public Security, Beijing 100038, China

**Keywords:** referring video object segmentation, cross-modal transformer, decoupled queries, feature pyramid network

## Abstract

Referring video object segmentation (R-VOS) is a fundamental vision-language task which aims to segment the target referred by language expression in all video frames. Existing query-based R-VOS methods have conducted in-depth exploration of the interaction and alignment between visual and linguistic features but fail to transfer the information of the two modalities to the query vector with balanced intensities. Furthermore, most of the traditional approaches suffer from severe information loss in the process of multi-scale feature fusion, resulting in inaccurate segmentation. In this paper, we propose DCT, an end-to-end decoupled cross-modal transformer for referring video object segmentation, to better utilize multi-modal and multi-scale information. Specifically, we first design a Language-Guided Visual Enhancement Module (LGVE) to transmit discriminative linguistic information to visual features of all levels, performing an initial filtering of irrelevant background regions. Then, we propose a decoupled transformer decoder, using a set of object queries to gather entity-related information from both visual and linguistic features independently, mitigating the attention bias caused by feature size differences. Finally, the Cross-layer Feature Pyramid Network (CFPN) is introduced to preserve more visual details by establishing direct cross-layer communication. Extensive experiments have been carried out on A2D-Sentences, JHMDB-Sentences and Ref-Youtube-VOS. The results show that DCT achieves competitive segmentation accuracy compared with the state-of-the-art methods.

## 1. Introduction

Referring video object segmentation (R-VOS) is an emerging subtask in the field of video segmentation. Its objective is to segment the regions of interest within video frames based on provided natural language expressions. In contrast to traditional semi-supervised video object segmentation methods [1,2,3], R-VOS only requires a simple textual description of the target object instead of manually annotating it in the first video frame (or a few frames). Consequently, R-VOS avoids intricate and costly annotation procedures, enhancing its user-friendliness. The academic community has recently exhibited great interest in this task due to its promising potential in applications such as intelligent surveillance video processing, video editing, and human–computer interaction.

Referring video segmentation addresses two crucial issues: how to make the model understand which object is the referred one, and how to accurately segment the target from the background.

The key to addressing the former issue lies in accomplishing fine-grained cross-modal information interaction and aggregation. Most of the early methods involve direct fusion of the visual and linguistic features. Common strategies include concatenation–convolution [4,5], recurrent LSTM [6], dynamic filter [7,8] and attention mechanism [9,10,11,12]. In contrast, some recent approaches [13,14,15,16] introduce additional variables as a bridge for aggregation. They employ a set of object queries to collect and store entity-related information from multi-modal features. Subsequently, these queries are linked across frames to achieve the tracking effect. Methods of this type can better model the cross-modal dependencies, but often overlook the scale disparity between visual and linguistic features. Since visual features are often tens of times longer than linguistic features, the object queries will be overly biased to focus on visual content.

The latter issue, which is to precisely identify the boundary between the reference object and background, is also important for improving segmentation accuracy. Most of the existing R-VOS methods [11,13,17] use a conventional Feature Pyramid Network (FPN) [18] to fuse features of multiple levels. Despite showing good performance, there is still a lot of room for improvement because of the information loss in this progressive fusion process.

To address the above problems, we propose DCT, an end-to-end decoupled cross-modal transformer for referring video object segmentation, which adopts a DETR-like [19] framework. In particular, in order to precisely distinguish the referred target, we firstly propose a Language-Guided Visual Enhancement Module (LGVE). It uses cross-modal attention operations to transmit discriminative linguistic information to visual features of all levels, so as to strengthen the response of the referred area and preliminarily filter out irrelevant background. In addition, a decoupled transformer decoder is designed. In this module, object queries interact with visual and linguistic features parallelly to reduce the attention bias caused by feature size differences. For accurate boundary locating, we leverage the existing Cross-layer Feature Pyramid Network (CFPN) [20] structure to conduct direct communication across multiple layers, which reduces the information loss during the commonly used stage-wise fusion process.

In summary, the main contributions of this paper are as follows:An end-to-end unified network termed DCT is proposed to tackle referring video object segmentation, which sufficiently utilizes multi-modal information and aggregates multi-scale visual features.The Language-Guided Visual Enhancement Module (LGVE) and the decoupled transformer decoder are constructed to establish coordinated information interactions among object queries, visual features and linguistic features.Cross-layer Feature Pyramid Network (CFPN) is brought in to reduce the information loss in the progressive fusion process.Experiments on four benchmarks demonstrate that our proposed method achieves competitive segmentation accuracy compared with the state-of-the-art methods.

## 2. Related Works

### 2.1. Referring Video Object Segmentation

Referring video object segmentation (R-VOS) is an important research area at the intersection of computer vision and natural language processing. Existing R-VOS methods can be broadly categorized into three types: propagation-based methods [5,12,21], matching-based methods [22,23], and query-based methods [13,14,16].

Propagation-based methods apply image-level referring object segmentation methods [9,24] on individual video frames and then acquire important temporal context through mask propagation. Seo et al. [12] designed a memory attention module based on self-attention architecture to propagate spatio- and temporal information from memory frames to the current frame, enhancing the temporal consistency of the segmentation results. Hui et al. [21] used textual information to guide the weighted combination of features from the current frame and reference frames, further optimizing the feature representation of the referred target. Propagation-based R-VOS methods are simple and fast, but prone to error accumulation, particularly when there are significant changes in the appearance of the target.

Matching-based methods divide the R-VOS task into two steps: trajectory generation and cross-modal matching. In the trajectory generation step, a model for instance segmentation or object detection is used to identify all the objects. Then, the objects are associated across the entire video to construct a collection of object trajectories. In the cross-modal matching step, the methods compute the relevance scores between each trajectory and the language description and select the pair which matches best. However, methods of this type [22,23] have a more complex training process as they require separate optimization for their multiple submodules.

Queried-based methods view R-VOS as a sequence prediction problem. They introduce a set of object queries to represent video entities and link them across frames to achieve natural tracking. Botach et al. [13] proposes the first query-based R-VOS method, which is named MTTR. The method includes no text-related inductive bias modules or post-processing operations, which greatly simplifies the segmentation process. However, unlike object detection and panoramic segmentation, object attributes in R-VOS are more random and difficult to accurately describe by fixed query vectors. In this regard, Wu et al. [14] take the given language expression as a constraint and generated query vectors online to make them more focused on the referred target.

### 2.2. Transformer

Transformer [25] is an attention-based encoder–decoder architecture which has a remarkable ability to capture long-term global dependencies. It was originally used for sequence modeling in machine translation and has been widely applied in natural language processing (NLP) [26,27] and computer vision (CV) [28,29] tasks. More recently, transformer has also been introduced to the highly regarded multi-modal domain, which provides valuable insights for R-VOS. For example, the large-scale pre-training model CLIP [30] uses transformers to extract visual and text features and accurately align them in the embedding space through contrastive learning. Ding et al. [31] exploits transformer as the cross-modal decoder for referring image segmentation and proposes a query generation module based on multi-head attention, which can comprehend the given language expression from different perspectives under the guidance of visual cues. The proposal of DETR [19], an end-to-end object detector, is a significant milestone in the development of transformer. It introduces the query-based paradigm and simplifies the conventional pipeline of object detection. MDETR [32] extends this idea to the field of referring expression comprehension, proposing an end-to-end modulated detector that detects objects in an image conditioned on a raw text query. VisTR [33] employs a non-auto-regressive transformer to parallelly supervise and segment the video instances at the sequence level. Considering the simplicity and efficiency of this DETR-like framework, our proposed method also adopts this architecture, but further addresses the undesirable attention bias in the interaction between object query and multi-modal features.

## 3. Method

### 3.1. Overall Pipeline

Given a T-frame video clip $V={\left\{v_{t}\right\}}_{t=1}^{T}$ with the spatial resolution of H×W and a L-word text expression $ℛ={\left\{r_{l}\right\}}_{l=1}^{L}$, the aim of DCT is to generate a binary segmentation mask sequence $ℳ={\left\{m_{t}\right\}}_{t=1}^{T}$ for the referred object. The overall pipeline of DCT is shown in Figure 1. It consists of four components: Feature Extraction and Enhancement, the decoupled transformer decoder, Instance Segmentation and Instance Sequence Matching process.

**Feature Extraction and Enhancement.** For the given video–text pair, we first use a visual encoder and a linguistic encoder for feature extraction, then use LGVE to achieve language-guided visual enhancement. Specifically, Video Swin transformer [34] is adopted to extract the multi-level visual features of the video frames, which are Fv1∈ℝT×C1×H4×W4, Fv2∈ℝT×C2×H8×W8 and Fv3∈ℝT×C3×H16×W16. Meanwhile, a pretrained language model RoBERTa [35] is employed to extract the word-level linguistic feature Fl∈ℝCl×L. Considering that the visual features do not include a particular focus on the referred object, we propose the LGVE to highlight language-related visual regions, generating the enhanced visual features Fei,i=1,2,3.

**Decoupled Transformer Decoder.** In this module, we introduce a set of N object queries Q={qt}t=1T, qt∈ℝN×Cq to represent the instances for each frame. Firstly, the top-level feature Fe3 output by LGVE is picked out and combined with fixed positional encodings. Then, we feed Q into stacked transformer decoder layers along with the features Fl and Fe3 to collect and store entity-related information. Details on this can be found in Section 4.3.

**Instance Segmentation.** On top of the transformer decoder and LGVE, we build a CFPN spatial decoder and two prediction heads to obtain the mask sequences. In particular, the CFPN spatial decoder takes the multi-level enhanced visual features as inputs and outputs the segmentation feature map $F_{seg}={\left\{f_{seg}^{t}\right\}}_{t=1}^{T}$ for each frame. The mask head comprises three stacked linear layers and produces dynamic kernels Ω from each object query, which is then convolved with $F_{seg}$ to obtain N instance sequences. The class head is a single-layer perceptron. It predicts the binary confidence score of each sequence, which indicates whether it matches the referred target.

**Instance Sequence Matching and Loss.** Having the instance sequences and their class scores, we proceed to find the optimal assignment between ground-truth and the predictions using Hungarian matching [36]. The loss of DCT is same as the one in [13], which is composed of ${ℒ}_{cls}$ and ${ℒ}_{mask}$. Specifically, ${ℒ}_{cls}$ is a cross-entropy loss while ${ℒ}_{mask}$ is a combination of the Dice [37] and binary Focal loss [38]. The whole loss function is as follows: (1)$$ℒ=\lambda_{cls}{ℒ}_{CE}+\lambda_{d}{ℒ}_{Dice}+\lambda_{f}{ℒ}_{Focal}$$
where $\lambda_{cls}$, $\lambda_{d}$ and $\lambda_{f}$ are three hyperparameters. More detailed settings can be seen in Section 4.2.

### 3.2. Language-Guided Visual Enhancement 

Since the visual features initially extracted by the backbone contain no special concentration on the referred object, it is important to convey the target-related linguistic semantics to redistribute their attention. MTTR [13] use a standard self-attention operation to facilitate information exchange only between the linguistic feature and the top-level visual feature, which may widen the semantic gap within the visual features of different levels. ReferFormer [14] proposes a Cross-modal Feature Pyramid Network to perform multi-scale cross-modal fusion but places it behind the transformer. As a result, the transformer still encounters difficulty in clearly identifying the target. Therefore, we design the Language-Guided Visual Enhancement Module, which incorporates linguistic information into visual features of all levels and puts it ahead. This module acts as a coarse locater and performs an initial filtering of irrelevant background regions.

The structure of LGVE is shown in Figure 2. Here, we use fv to represent the multi-level visual features of a single frame, and fvi∈ℝCi×Wi×Hi is the one of level-i. Firstly, our LGVE generates the query (Q) and the intermediate visual feature fmidi from fvi with 1 × 1 point-wise convolutions followed by 3 × 3 depth-wise convolutions, aggregating pixel-wise cross-channel context and channel-wise spatial context. Simultaneously, the key (K) and value (V) are generated from the linguistic feature Fl with two linear projections. Then, we use cross-attention operations to assemble word-level linguistic features at each spatial location and produce a vision–language correlation filter S, whose dimension is the same as fvi. The above process is shown in Equations (2) and (3).
(2)(fmidi,Q,K,V)=(WdmWpmfvi,WdqWpqfvi,WlkFl,WlvFl)
(3)S=Softmax(QKTCi)V
where Wd(⋅), Wp(⋅) and Wl(⋅) are 1 × 1 point-wise convolution, 3 × 3 depth-wise convolution and linear projections, respectively.

Finally, the intermediate feature $f_{mid}^{i}$ is multiplied by the spatial filter *S* element-wisely to obtain the i-th level enhanced visual feature $f_{e}^{i}$:(4)$$f_{e}^{i}=S⊙f_{mid}^{i}$$

### 3.3. Decoupled Transformer Decoder

The transformer module is a key component in query-based R-VOS methods. Its objective is to gather entity-related information from the multi-modal features and store it in the object queries. However, the decoding process of existing works ignores the severe imbalance in the sizes of multi-modal features. As the length of the visual feature is much longer than that of the linguistic feature (often 20 times longer or more), the object queries may be greatly biased to the former, which is not conducive to a fine-grained understanding of the language expression.

To address this problem, we consider separately interacting the object queries with features of the two modalities and merging the results with adaptive weights. As is shown in Figure 3, the decoupled transformer decoder has $N_{d}$ layers. In each decoder layer, we first perform a self-attention operation upon the object queries *q* to model its inner relationship and output $q_{s}$. Then, two cross-attention layers are constructed to collect information from the linguistic feature $F_{l}$ and enhanced visual feature $f_{e}^{3}$ independently, generating the single-model subqueries $q_{text}$ and $q_{video}$. Finally, the updated query $q^{\prime }$ is obtained by a weighted sum of the subqueries, where the weights are learned from the subquery embeddings with liner projections. The above process can be represented by Equations (5)–(7).
(5)$$q_{s}=\mathrm{LN}(\mathrm{MSA}(q)+q)$$
(6)$$q_{text}=\mathrm{LN}(\mathrm{MCA}(q_{s},F_{l})+q_{s})\;q_{video}=\mathrm{LN}(\mathrm{MCA}(q_{s},f_{e}^{3})+q_{s})$$
(7)$$q^{\prime }=q_{text}\times \theta_{l}(q_{text})+q_{video}\times \theta_{v}(q_{text})$$
where MSA() and MCA() are multi-head self-attention and multi-head cross-attention. LN represents layer normalization. $\theta_{l}$ and $\theta_{v}$ are fully connected layers.

### 3.4. Cross-Layer Feature Pyramid Network

As described earlier, FPN is a classical method for fusing multi-scale features in R-VOS methods and brings about obvious improvements in most cases. However, in this kind of progressive fusion process, the low-level visual information (e.g., object texture and boundary details) is accessed only once in the final fusion stage, resulting in a low-qualified segmentation of the edges. Additionally, as the fusion proceeds, high-level semantic cues are gradually diluted, which diminishes the model’s ability to recognize the referred target. In view of the above two issues, we replace standard FPN with the Cross-layer Feature Pyramid Network (CFPN) [20]. Compared with FPN, CFPN promotes the information exchange among visual features of all layers by aggregating them simultaneously, thus generating a segmentation feature map rich in both semantics and spatial details.

As shown in Figure 4, CFPN first performs global average pooling (GAP) and concatenation operations on the enhanced visual features, resulting in a 1d global representation, *Z*. Then, we transform *Z* into the layer-wise fusion weight ψ={φi}i=13 with a two-layer perceptron, formulated as
(8)ψ=θ1(ReLU(θ2(Z)))
where θ1∈ℝD×Y and θ2∈ℝY×3 are fully connected layers, D=C1+C2+C3 is the channel number of *Z*, *Y* is set to 256 empirically, and ReLU refers to the ReLU activation function. Afterwards, the dynamical fusion weight is used to rescale the original visual features and form the aggregated visual representation fg
(9)fg=(fe1×φ1)⊕UP(fe2×φ2)⊕UP(fe3×φ3)
where ⊕ denotes the concatenation operation and UP refers to upsampling. Considering that fg is a naïve concatenation of the multi-level features, CFPN further constructs a cross-layer feature distribution structure. To be more precise, fg is fed into a set of average pooling layers followed by 3 × 3 convolutions to generate the redistributed features fdi, i=1,2,3, which are subsequently merged in a top–down manner to yield the final segmentation feature map fseg. In contrast to FPN, the feature maps in fd collect information from the full spectrum of multi-level representation. This enables the retention of more discriminative and complementary visual information during the fusion process.

## 4. Experiments

### 4.1. Datasets and Evaluation Metric

#### 4.1.1. Datasets

To evaluate the proposed method, we experiment on three open-sourced R-VOS datasets: A2D-Sentences [7], JHMDB-Sentences [7] and Ref-Youtube-VOS [12].

**A2D-Sentences** is an extension of the A2D [39] dataset. It consists of 3782 YouTube videos and 6655 textual descriptions, covering eight types of actions performed by seven categories of objects.

**JHMDB-Sentences** is an extension of the J-HMDB [40] dataset. It comprises 928 video sequences showcasing 21 human actions, each accompanied by a corresponding textual description. Notably, each frame of the videos is labeled with a 2d puppet mask.

**Refer-YouTube-VOS** is built upon the large-scale video segmentation dataset YouTube-VOS [41]. It consists of a total of 3978 videos, of which 3471 are used for training, 202 for validation, and 305 for testing. Each video in this dataset is annotated with high-quality instance segmentation masks for every fifth frame. Since the test set is accessible only during the competition, our evaluation experiments are conducted on the validation set.

#### 4.1.2. Evaluation Metric

For A2D-Sentences and JHMDB-Sentences, we adopt precision at thresholds of 0.5, 0.6, 0.7, 0.8, and 0.9(P@X), overall Intersection-over-Union (OIoU), mean Intersection-over-Union (MIoU) and mean average precision over 0.50:0.05:0.95 (mAP) as evaluation metrics. 

For Refer-YouTube-VOS, the method is evaluated with the criteria of region similarity (J), contour accuracy (ℱ) and their average value (J&ℱ).

### 4.2. Inplementation Details

In accordance with previous works [13,14], we train DCT on A2D-Sentences and Refer-YouTube-VOS and use all three datasets for evaluation. For model settings, the decoupled transformer has 4 decoder layers ($N_{d}=4$) and each layer is configured with 8 attention heads. The number of object queries is set to 50. The hyperparameters of the loss function are set as $\lambda_{cls}=2$, $\lambda_{d}=5$ and $\lambda_{f}=2$.

During training, we use sliding windows to crop video clips and the default window size is set to eight. The resolution of the frames is adjusted to ensure that the shorter side is at least 360 pixels and the longer side is at most 640 pixels. Random horizontal flipping, random cropping, and photometric distortion are used for data augmentation. AdamW is used as the optimizer and the weight decay is set to 1 × 10^−4^. For A2D-Sentences, we train the model for 60 epochs with a batch size of 6 and a dynamic learning rate, which is as shown in Equation (10). For Refer-YouTube-VOS, the epoch and batch size are 30 and 4, respectively, and the learning rate is shown in Equation (11).
(10)$$lr_{A2D}(\mathrm{epoch})=\left\{ \begin{array}{l} 0.0001\;,0\leq \mathrm{epoch}<40\\ 0.00005\;,40\leq \mathrm{epoch}<60 \end{array}\right.$$
(11)$$lr_{\mathrm{Refer}}(\mathrm{epoch})=\left\{ \begin{array}{l} 0.0001\;,0\leq \mathrm{epoch}<20\\ 0.00005\;,20\leq \mathrm{epoch}<30 \end{array}\right.$$

When inferencing, DCT predicts *N* instance sequences corresponding to the *N* queries. For each sequence, we sum the confidence scores output by the class head across frames. Finally, the sequence with the highest total score is identified as the referred object.

### 4.3. Ablation Study

#### 4.3.1. Ablation Study on the Main Components

We conduct extensive experiments on A2D-Sentences to evaluate the effectiveness of the key components in our proposed method. The results are presented in Table 1. In the baseline model (as shown in the first line), the LGVE is removed while the decoupled transformer decoder and CFPN are replaced with a conventional transformer decoder and a common FPN similar to the ones used in MTTR [13]. In this case, the OIoU, MIoU and mAP drop significantly by 3.8, 4.1 and 5.4. In experiment No.2 to No.4, we introduce the three proposed components on top of the baseline separately. Obviously, each component can bring about an improvement in the segmentation accuracy. In experiment No.5, we achieve a higher accuracy by simultaneously introducing the LGVE and CFPN. The reason for this may be that the LGVE facilitates the semantic consistency of the visual features, thus reducing the misalignment caused by semantic discrepancies during the global fusion in CFPN.

#### 4.3.2. Analysis of the Temporal Context Size

Modeling the temporal context plays a crucial role in the R-VOS task for which actions or behaviors in videos cannot be fully understood or derived by analyzing a single frame. In DCT, we use sliding windows to crop video clips and adopt Video Swin transformer as the spatial–temporal encoder. To study the effect of the temporal context size, we change the window size during training and evaluating on A2D-Sentences dataset. The results are presented in Table 2. When the window size is 1, the model essentially transforms into an image-level approach, with mAP being only 42.6. As the window size increases, the model captures more time clues, and the segmentation performance gradually improves. However, the metrics reach their peak at the window size of 8. One possible reason for this phenomenon could be that over a longer time span, the target’s behavior and spatial position undergo more pronounced changes, which consequently make the target more challenging for the model to comprehend.

#### 4.3.3. Analysis of the Query Number

To investigate the impact of the number of object queries on model performance, we conduct ablation experiments on the A2D-Sentences dataset under the window size of 8, as shown in Table 3. During the training process, randomly initialized object queries gradually converge towards fixed regions or specific categories of targets. As a result, a low number of queries (e.g., 5) are insufficient to cover the complex distribution of objects in the dataset. On the other hand, when queries are too dense (e.g., 75), more similar mask sequences are generated, making it more challenging for the model to optimize during the one-to-one Hungarian matching. Therefore, the number of queries for the final model is set to 50.

### 4.4. Comparison with Existing Methods

**Results on A2D-Sentences and JHMDB-Sentences.** We compare our proposed method on A2D-Sentences and JHMDB-Sentences with the current state-of-the-art methods, including Hu et al. [4], Gavrilyuk et al. [7], CMSA + CFSA [42], ACAN [43], CMPC-V [44], ClawCraneNet [45], MTTR [13] and ReferFormer [14]. As shown in Table 4, our proposed method achieves 73.5 OIoU, 65.0 MIoU and 48.7 mAP on A2D-Sentences, which greatly outperforms previous methods using CNN-based visual encoders (e.g., CMPC-V and ClawCraneNet). Besides, when equally employing Video-Swin-T as the visual encoder, the proposed method also has certain improvements over the query-based methods MTTR and ReferFormer. Table 5 shows the performance comparison on JHMDB-Sentences dataset. It can be seen that DCT achieves OIoU, MIoU and mAP values of 70.7, 70.5 and 39.6, respectively, which are 0.7, 1.2 and 0.5 higher than the best method ReferFormer. As the results on JHMDB-Sentences are obtained by evaluating the model trained on A2D-Sentences without finetuning, DCT further proves its good generalization.

**Results on Refer-YouTube-VOS.** Table 6 displays the experimental results on the largest and challenging R-VOS dataset Refer-YouTube-VOS. For fair comparison, we report the performance of ReferFormer trained from scratch and without post-process operations. It can be observed that our DCT surpasses all the cutting-edge methods with a gain of 0.6J and 0.5ℱ and achieves the state of the art.

### 4.5. Visualization and Analysis

In Figure 5, we show some of the visualization results of DCT on Ref-Youtube-VOS. It can be observed that our DCT can accurately comprehend the given language expression and accomplish precise segmentation even in cases of severe object deformation (as shown in row 1), interference from similar objects (as shown in row 2) and partial disappearance of the target (as shown in row 3).

## 5. Discussion

In this paper, we optimize the modules of multi-modal feature interaction and decoding segmentation specially, which brings about a great improvement in the segmentation performance. However, the method still has some limitations. For example, we link the masks across frames according to the permutation of the queries, but do not design a memory module to store the historical features of the referred targets, resulting in an insufficient utilization of the temporal information. In addition, during the inference process, we select the mask sequence with the highest total confidence score as the final output. This approach may lead to false-positive results when the referred target does not appear in the video. In summary, how to collect and utilize temporal information from historical frames while maintaining the simplicity of the model framework, as well as improving the strategy for instance matching, remains worthy of further research.

## 6. Conclusions

This paper presents an end-to-end decoupled cross-modal transformer for referring video object segmentation. In the proposed model, a Language-Guided Visual Enhancement Module (LGVE) and a decoupled transformer decoder are designed to establish sufficient and balanced information interactions between the object queries and features of different modalities, so as to accurately identify the referred object. Then, we introduce the Cross-layer Feature Pyramid Network, which was originally used for salient object detection, as the spatial decoder, generating high-qualified object boundaries by making a better use of the visual semantics and details. Extensive experiments are carried out on three benchmark datasets. The experimental results fully demonstrate the effectiveness of the proposed method.

## Figures and Tables

**Figure 1 sensors-24-05375-f001:**
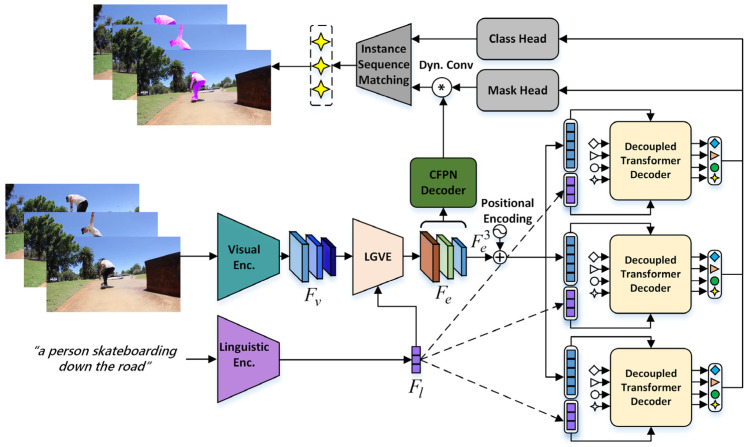
Overall architecture of the proposed method.

**Figure 2 sensors-24-05375-f002:**
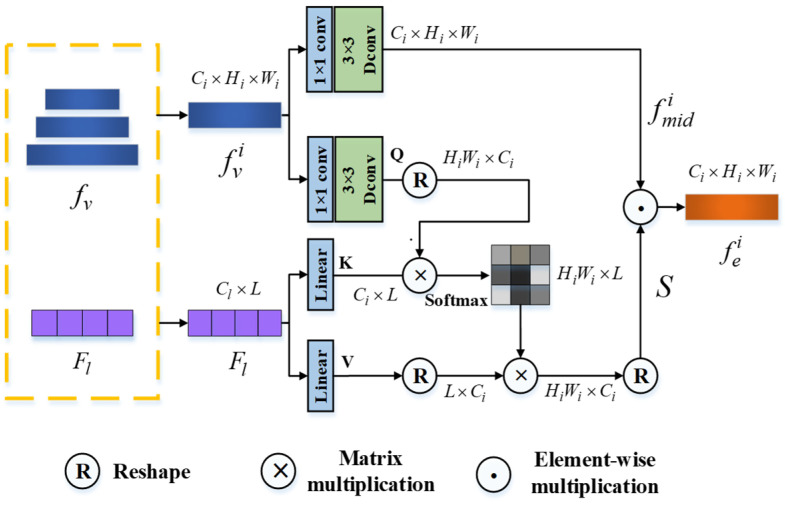
Language-Guided Visual Enhancement Module.

**Figure 3 sensors-24-05375-f003:**
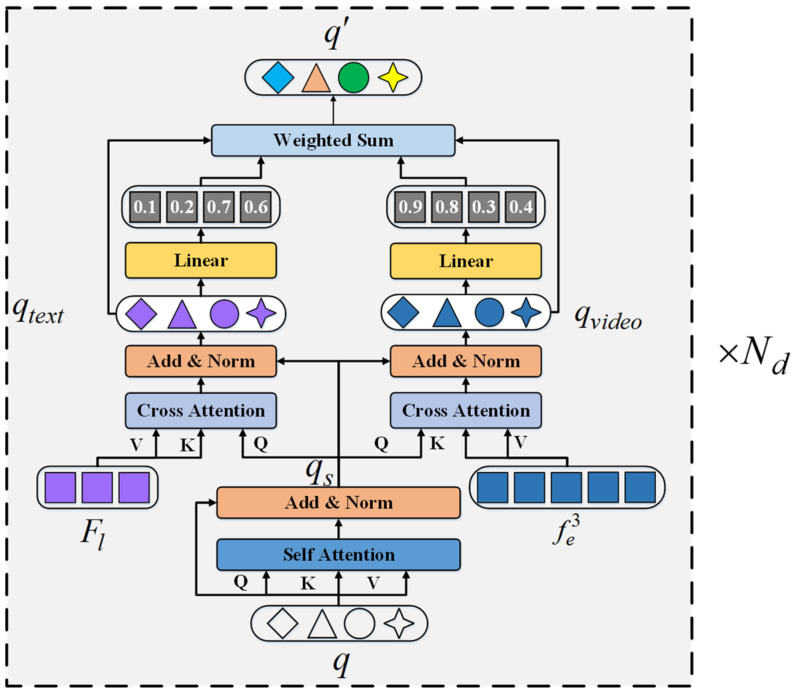
Decoupled transformer decoder.

**Figure 4 sensors-24-05375-f004:**
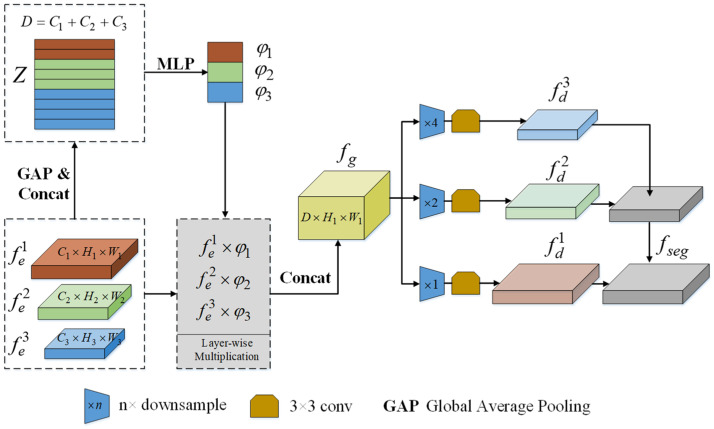
Cross-layer Feature Pyramid Network.

**Figure 5 sensors-24-05375-f005:**
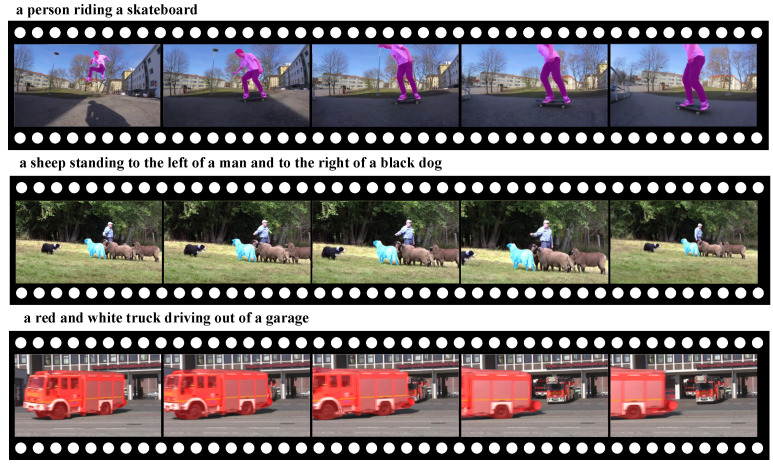
Visualization results on Ref-Youtube-VOS dataset.

**Table 1 sensors-24-05375-t001:** Ablation experiments on the key components.

No.	LGVE	Decoupled Transformer Decoder	CFPN	IoU		mAP
				Overall	Mean	
1	-	-	-	69.7	60.9	43.3
2	√	-	-	71.8	63.5	46.3
3	-	√	-	71.0	62.7	45.1
4	-	-	√	72.3	63.3	46.8
5	√	-	√	73.0	64.2	47.2
6	√	√	√	**73.5**	**65.0**	**48.7**

**Table 2 sensors-24-05375-t002:** Ablation experiments on the sliding-window size.

Window Size	IoU		mAP
	Overall	Mean	
1	69.7	61.0	42.6
4	70.8	62.0	44.5
6	71.9	63.3	46.1
8	**73.5**	**65.0**	**48.7**
10	72.1	63.8	46.5

**Table 3 sensors-24-05375-t003:** Ablation experiments on the query number.

Query Number	IoU		mAP
	Overall	Mean	
5	71.8	63.0	45.8
30	73.1	64.4	47.5
50	**73.5**	**65.0**	**48.7**
75	72.3	63.6	46.4

**Table 4 sensors-24-05375-t004:** Performance comparison on A2D-Sentences dataset.

Method	Backbone	Precision					IoU		mAP
		P@0.5	P@0.6	P@0.7	P@0.8	P@0.9	Overall	Mean	
Hu et al. [4]	VGG-16	34.8	23.6	13.3	3.3	0.1	47.4	35.0	13.2
Gavrilyuk et al. [7]	I3D	47.5	34.7	21.1	8.0	0.2	53.6	42.1	19.8
CMSA + CFSA [42]	ResNet-101	48.7	43.1	35.8	23.1	5.2	61.8	43.2	-
ACAN [43]	I3D	55.7	45.9	31.9	16.0	2.0	60.1	49.0	27.4
CMPC-V [44]	I3D	65.5	59.2	50.6	34.2	9.8	65.3	57.3	40.4
ClawCraneNet [45]	ResNet101	70.4	67.7	61.7	48.9	17.1	63.1	59.9	-
MTTR [13]	Video-Swin-T	75.4	71.2	63.8	48.5	16.9	72.0	64.0	46.1
ReferFormer [14]	Video-Swin-T	76.0	72.2	65.4	49.8	17.9	72.3	64.1	48.6
DCT (ours)	Video-Swin-T	76.3	72.8	66.0	50.2	18.3	73.5	65.0	48.7

**Table 5 sensors-24-05375-t005:** Performance comparison on JHMDB-Sentences dataset.

Method	Backbone	Precision					IoU		mAP
		P@0.5	P@0.6	P@0.7	P@0.8	P@0.9	Overall	Mean	
Hu et al. [4]	VGG-16	63.3	35.0	8.5	0.2	0.0	54.6	52.8	17.8
Gavrilyuk et al. [7]	I3D	69.9	46.0	17.3	1.4	0.0	54.1	54.2	23.3
CMSA + CFSA [42]	ResNet-101	76.4	62.5	38.9	9.0	0.1	62.8	58.1	-
ACAN [43]	I3D	75.6	56.4	28.7	3.4	0.0	57.6	58.4	28.9
CMPC-V [44]	I3D	81.3	65.7	37.1	7.0	0.0	61.6	61.7	34.2
ClawCraneNet [45]	ResNet101	88.0	79.6	56.6	14.7	0.2	64.4	65.6	-
MTTR [13]	Video-Swin-T	93.9	85.2	61.6	16.6	0.1	70.1	69.8	39.2
ReferFormer [14]	Video-Swin-T	93.3	84.2	61.4	16.4	0.3	70.0	69.3	39.1
DCT (ours)	Video-Swin-T	94.7	85.5	62.0	16.9	0.1	70.7	70.5	39.6

**Table 6 sensors-24-05375-t006:** Performance comparison on Refer-YouTube-VOS dataset.

Method	Backbone	J&ℱ	J	ℱ
CMSA [9]	ResNet-50	34.9	33.3	36.5
URVOS [12]	ResNet-50	47.3	45.3	49.2
PMINet [46]	ResNeSt-101	48.2	46.7	49.6
MTTR [13]	Video-Swin-T	55.3	54.0	56.6
ReferFormer [14]	Video-Swin-T	56.1	54.8	57.3
DCT (ours)	Video-Swin-T	**56.6**	**55.4**	**57.8**

## Data Availability

The data that support the findings of this study are available online. These datasets were derived from the following public resources: [A2D-Sentences, JHMDB-Sentences, Ref-Youtube-VOS].

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
