# Peer review of "Decoupled Cross-Modal Transformer for Referring Video Object Segmentation"

_sensors, 2024, doi:10.3390/s24165375_

Round 1

Reviewer 1 Report

Comments and Suggestions for Authors

The paper is well structured. However, some major revisions are needed:
The Matching-based methods section should cite relevant literature.
It would be helpful to include a discussion on the method's limitations and possible future improvements.

Author Response

Comments 1: The Matching-based methods section should cite relevant literature.

Response 1: We have added citations at section 2.1 in the revised manuscript. (Line 107)

Comments 2: It would be helpful to include a discussion on the method's limitations and possible future improvements.

Response 2: Thank you for your professional advice. We have added a Discussion section containing the method's limitations and possible future improvements. (Line 383-Line 394)

Reviewer 2 Report

Comments and Suggestions for Authors

This paper proposes a method referred to as a decoupled cross-model transformer for the task of referring video object segmentation.

Proposed DCT is composed of three modules; (a) Language Guided Visual Enhancement Module, (b) Decoupled Transformer Decoder, and (c) Cross-layer Feature Pyramid Network.

- Third paragraph of section 2.1 should contain references to existing works. 

- MTTR, ReferFormer methods should be cited in L 186, 189

- Figures should be center aligned.

- The proposed work seems a little bit old-fashioned. Utilizing LLM with multi-modal interaction (e.g., Q-former) might benefit the performance and novelty of proposed method.

Author Response

Comments 1: Third paragraph of section 2.1 should contain references to existing works.

Response 1: We have added citations at section 2.1 in the revised manuscript. (Line 107)

Comments 2: MTTR, ReferFormer methods should be cited in L 186, 189

Response 2: The citations are added in the revised manuscript. (Line 186,189)

Comments 3: Figures should be center aligned.

Response 3: Thank you for pointing this out and we have revised it according to the journal format.

Comments 4: The proposed work seems a little bit old-fashioned. Utilizing LLM with multi-modal interaction (e.g., Q-former) might benefit the performance and novelty of proposed method.

Response 4: Thank you very much for your professional advise. We will conduct relevant researches and explorations about LLM in the future.
